

# Clinical, molecular, and resistance features of *Listeria monocytogenes* in non-perinatal patients with listeriosis: 8-year retrospective data from four tertiary hospitals in Shandong, China

Yan Liu[1],[*], Yuan Xie[1],[*], Lei Liu[2], Jie Wang[1], Wenjing Li[3], Chengfan Yang[4] and Shuhua Lu[1]

[1] Clinical Laboratory, Affiliated Hospital of Jining Medical University, Jining, Shandong, China
[2] Shandong Provincial Center for Disease Control and Prevention, Jinan, Shandong, China
[3] Institute of Cerebrovascular Disease, Xianyang Hospital of Yan'an University, Xianyang, China
[4] Clinical Medicine, Jining Medical University, Jining, Shandong, China
[*] These authors contributed equally to this work.

Corresponding author
Shuhua Lu,
lushuhua010515@126.com

## ABSTRACT

**Background:** Listeriosis, caused by *Listeria monocytogenes* (*L. monocytogenes*), is a severe infectious disease with high hospitalization and fatality rates. Urgent epidemiological studies on this disease with geographic variability are needed, particularly in developing countries.

**Methods:** This study included non-perinatal patients with listeriosis from four tertiary centers in Shandong, China. Data on demographics, clinical characteristics, and outcomes were collected retrospectively over 8 years (2015–2022).

**Results:** Among 292,254 non-perinatal patients, 27 listeriosis cases were identified, resulting in an incidence of nine cases per 100,000 admissions. Patients had a median age of 56 years, with 66.7% having comorbidities. Fever was the most common symptom (88.9%), and 44.4% had central nervous system involvement. Most patients (26/24; 96.3%) received antibiotics, 37.0% received monotherapy and 59.3% combination therapy. The mortality rate was 22.2%. The antimicrobial susceptibility test showed that 25 strains were sensitive to most antibiotics. Multilocus sequence typing revealed nine sequence types (ST), predominantly ST8 (44.4%), and serogroup 1/2a is the most common serogroup (66.7%).

**Conclusions:** This study provides valuable insights into the clinical and molecular features of *L. monocytogenes* in non-perinatal patients with listeriosis. The findings demonstrated the predominance of serogroup 1/2a and ST8. Despite low resistance and appropriate treatment, listeriosis remains associated with a significant mortality rate, emphasizing the need for timely diagnosis and effective management strategies.

## INTRODUCTION

Listeriosis, caused by the facultative anaerobic *Listeria monocytogenes* (*L. monocytogenes*), is a severe foodborne infectious disease, commonly resulting in high hospitalization and

fatality rates approaching 50% (*de Noordhout et al., 2014*). Invasive infections can cause a series of severe clinical outcomes (*e.g.*, meningitis, bacteremia, and meningoencephalitis) and are even life-threatening, particularly in the elderly or immunocompromised hosts (*Schlech, 2019*). Although listeriosis usually occurs infrequently and sporadically, outbreaks are reported occasionally (*Papic et al., 2020*). Because of its low incidence, the clinical evidence of human listeriosis is limited (*Koopmans et al., 2023*), without a comprehensive analysis of both epidemiological and microbiological data. In addition, the excessive use of antibiotics leads to the drug resistance of the pathogen *L. monocytogene*, aggravating the health burden and the concerns about the management of this disease (*Maćkiw et al., 2016*). Therefore, the investigation of listeriosis and pathogens is of great importance to develop appropriate prevention and disease management strategies (*Pettinati, Telles & de Carvalho Ballian, 2006*).

Over the past decades, the incidence of listeriosis has been increasing worldwide (*Corral et al., 2024*; *Vazquez et al., 2024*; *Zahid et al., 2023*). However, epidemiological surveillance of listeriosis was not widely initiated in China until 2013 (*Lu et al., 2021*). Accordingly, the limited reports on Chinese cases mostly come from regional and historical data, leading to the underrepresentation of the overall Chinese population (*Feng et al., 2013*; *Zhang et al., 2019*). The available data on molecular features, such as serotypes, virulence genes, and multilocus sequence typing (MLST), is even more limited (*Fan et al., 2019*). Thus, no specific diagnosis and treatment consensus for this disease has been established in China up to now (*Lu et al., 2021*); and meanwhile, the impact of demographic and clinical characteristics on prognosis has not been fully assessed. Within this context, urgent efforts are needed to gather more data to better understand the factors affecting treatment and prognosis. Besides, previous reports have demonstrated that *L. monocytogenes* has a high strain heterogeneity in terms of virulence potential and environmental adaption (*Lakicevic, Den Besten & De Biase, 2021*), which underscores the necessity of additional analysis in other regions to fill the missing data in this large population.

Herein, this study represented the first analysis of the clinical features, molecular characteristics, and antimicrobial resistance profiles of *L. monocytogenes* isolates in non-perinatal patients with listeriosis over 8 years in Shandong Province, China, aiming to provide additional evidence regarding the heterogeneity of this species with geographic variability.

## MATERIALS AND METHODS

### Patient selection and data collection

In this multicenter, retrospective observational study, all non-perinatal patients with a microbiological diagnosis of listeriosis were included by reviewing the data from four tertiary centers in Shandong Province, China, between January 2015 and December 2022. Patients with culture-proven listeriosis, regardless of age or sex, were eligible to be included in this analysis. A patient was considered to be culture positive when *L. monocytogenes* was isolated from sterile clinical specimens (*e.g.*, cerebrospinal fluid (CSF), blood, ascites, joint fluids) in clinical microbiology laboratories, according to the previous definition (*Shi et al., 2021*).

Cases with pregnancy-associated listeriosis (including pregnant women, fetuses, or neonates) were not included.

Clinical and outcome data of the identified patients were retrieved from the electronic medical records, including the key demographic characteristics (age, sex, *etc.*), clinical symptoms (fever, abdominal pain, diarrhea, *etc.*), comorbidities (malignancy, autoimmune disease, *etc.*), laboratory results, treatment type, and clinical outcomes (cure, improvement or death). Direct communication with attending doctors was required for the missing data or further clarification. Investigators cross-checked the cases and information. This study was approved by the Ethics Committee of the Affiliated Hospital of Jining Medical University (No. 2022-08-C004). Due to the retrospective nature of the study, informed consent was waived by the ethics committee.

## Bacterial isolates and identification

Bacterial identification of isolates was performed based on the standard microbiological culture. After the collection and grind of clinical specimens, specimens were inoculated in a Columbia blood agar plate and stained with Gram stain. The CAMP testing was performed with a β-hemolytic strain of *Staphylococcus aureus* (ATCC 25923) and *Streptococcus agalactiae*. According to the morphological characteristics, Gram-positive, facultative anaerobes, Christie Atkins Munch-Peterson (CAMP)-positive, translucent colonies that produced a narrow β-hemolysis were presumptively identified as *L. monocytogenes* (Figs. 1A–1D). Further identification was confirmed by the automated Vitek-2 Compact system (bioMérieux, Lyons, France) with confidence levels of 99.9% for all strains and Vitek mass spectrometry system (bioMérieux, Lyons, France) with confidence levels of 98% (Figs. 1E, 1F).

## Antimicrobial susceptibility tests

The antimicrobial susceptibility tests (AST) was conducted using the Kirby-Bauer disk diffusion (KB) method on the automated Vitek-2 Compact system, following the recommendations of the Clinical and Laboratory Standards Institute (CLSI) guidelines. The tested antibiotics were selected according to the clinical experience and investigator selection, including erythromycin, ciprofloxacin, levofloxacin, gentamicin, meropenem, imipenem, cotrimoxazole, vancomycin, penicillin, ampicillin, and amikacin. The minimum inhibitory concentrations (MICs) of each antibiotic were measured for the interpretation of antibiotic susceptibility. The antibiotic susceptibility was judged as susceptible, intermediate, or resistant based on the breakpoints recommended in CLSI M45 3rd edition.

## Evaluation of molecular features

For the serogroup determination of *L. monocytogenes* isolates, the multiplex PCR assay was conducted using the following target genes: *prs*, *lmo0737*, *lmo1118*, *ORF2110*, and *ORF2819*, according to the method described by *Doumith et al. (2004)*. As per the amplifications of the four chosen serovar-specific fragments, *L. monocytogenes* strains were separated into four serogroups (1/2a, 1/2b, 1/2c, and 4b). A total of 11 virulence genes were

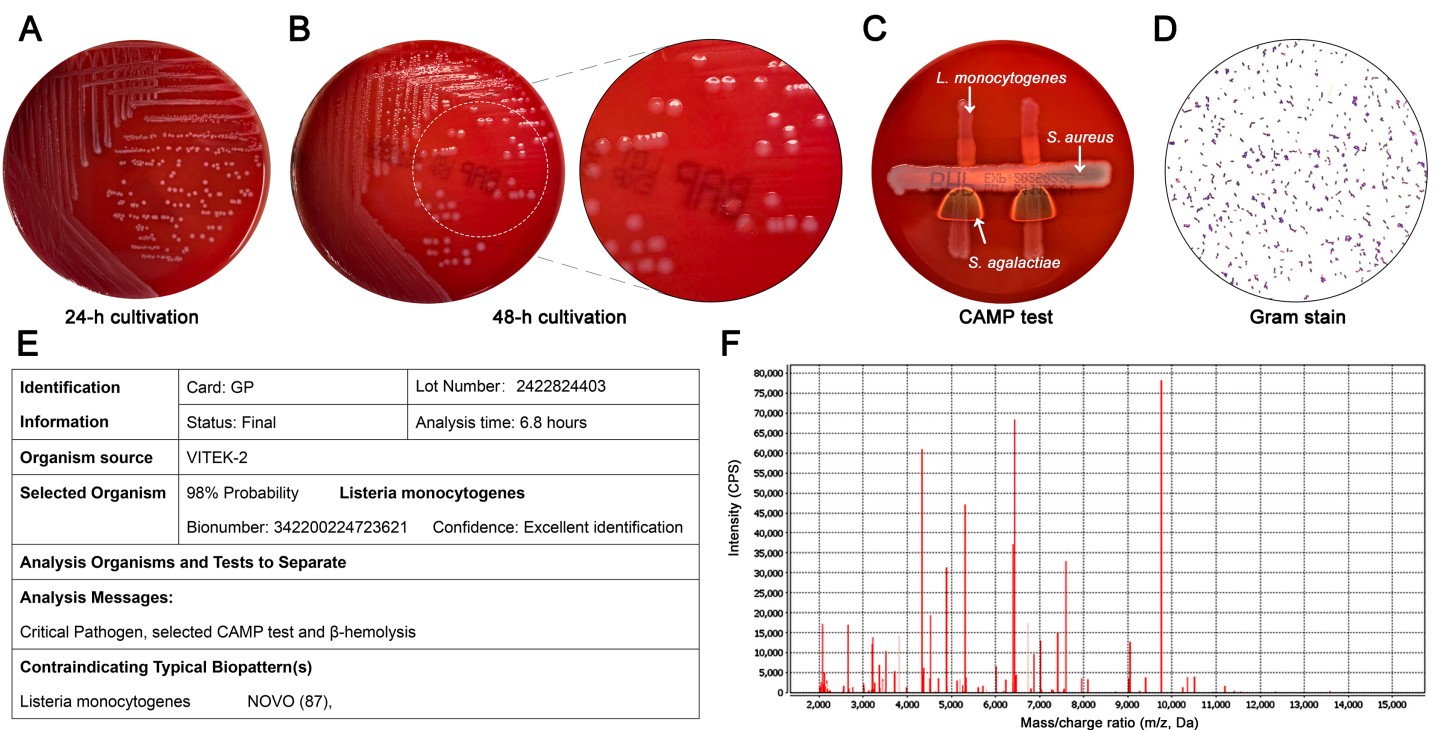

**Figure 1** **Microbiological features of *L. monocytogenes* isolates.** (A) Morphology of colonies at 24 h. Non-hemolytic, translucent and small white colonies were shown; (B) Morphology of colonies at 48 h. A narrow zone of β-hemolysis appeared on Columbia blood agar; (C) CAMP test of *L. monocytogenes* (top) with *S. aureus* (middle) showed the enhanced zone of hemolysis (synergistic hemolysis) towards *S. aureus*; (D) Gram stain showing Gram-positive colonies (magnification, ×100); (E) Representative result of Vitek-2 Compact system analysis (confidence levels, 99.9%). (B) Representative result of Vitek mass spectrometry analysis (confidence levels, 98%).

also detected with the multiplex PCR assay, including *hly*, *plcA*, *plcB*, *prfA*, *mpL*, *actA*, *inlA*, *inlB*, *inlC*, *inlJ*, and *iap*. All primers for the detection of serogroup and virulence genes are presented in Table S1.

As per the established scheme by *Ragon et al. (2008)*, MLST of all isolates was performed by amplifying and sequencing the internal fragments of seven housekeeping genes (*abcZ*, *bglA*, cat, *dapE*, *dat*, *ldh*, *lhkA*). The alleles and sequence types (STs) were determined by comparison with allelic profiles for *L. monocytogenes* in the MLST database (https://bigsdb.pasteur.fr/listeria/).

## Statistical analysis

Characteristics and outcomes of included patients were descriptively summarized as median (range) for continuous variables and frequency (percentage) for categorical variables. The incidence was calculated as the number of non-perinatal patients with diagnosed listeriosis divided by the number of the total population of non-perinatal patients admitted in the four centers over the study years. All analyses were performed by SPSS software version 27.0 (IBM Corp., Armonk, NY, USA).

## RESULTS

### Incidence of listeriosis

Throughout the study period, a total of 27 cases with listeriosis were identified from 292,254 non-perinatal patients at four centers, with an overall incidence of nine cases per 100,000 admissions. The number of identified cases and annual incidence of listeriosis per 100,000 admissions over the study period is provided in Fig. 1. The annual incidence generally increased from 6.3 to 17.6 cases per 100,000 admissions between 2015 and 2019, and there was a large drop since 2020 (Fig. 2).

### Clinical characteristics of patients with listeriosis

The majority of included patients were female (55.6%). All patients had a median age of 56 years (range, 8–82), including five (18.5%) patients aged <18 years and 11 (40.7%) patients aged ≥60 years (Table 1). A total of 18 (66.7%) patients had at least one comorbidity, mainly including hypertension, diabetes mellitus, malignancy, and autoimmune disease. The most common clinical symptom at presentation was fever (24/27 [88.9]). Most *L. monocytogenes* strains were isolated in the blood cultures of 17 patients (63.0%). There were 12 patients (44.4%) with central nervous system (CNS) involvement and two (7.4%) patients with other localized infections (one each ascites and joint fluid).

### Treatment and outcomes of patients with listeriosis

Of 27 analyzed patients, only one (3.7%) patient did not receive antibiotics therapy and 26 (96.3%) patients received antibiotics, including 10 (37.0%) with monotherapy and 16 (59.3%) with combination therapy. Among 10 patients with monotherapy, three received meropenem, three ampicillin, two imipenem-cilastatin, and two piperacillin-tazobactam. The combination therapies used are varied among these patients. Details of all patients' characteristics, treatments, and outcomes are provided in Table S2. The treatment duration varied from 1 day to 20 days. Among the 26 treated patients, 11 patients were cured (seven with monotherapy and four with combination therapy), nine improved (three *vs*. three), and six died (three *vs*. three) after appropriate treatments. Among 11 cured patients, seven have been administrated with penicillins, four with carbapenems, and two with cephalosporins. Among six patients who died, five were treated with carbapenems, three with cephalosporins, and two with penicillins.

### Antimicrobial resistance profile and molecular features of *L. monocytogenes* isolates

From the antimicrobial susceptibility tests, only two strains were identified to be resistant to erythromycin or ciprofloxacin (one for each strain). The remaining 25 strains were all sensitive to the tested antibiotics (Table 2). No strains were identified to be multidrug-resistant strains.

Of the 27 clinical isolates, nine (33.3%) isolates belonged to lineage I and 18 (66.7%) to lineage II. Nine sequence types (STs) were distinguished by MLST and each ST belonged to the same serotype (Table 3). The ST8 (44.4%) was the most commonly identified type, followed by the ST87 (14.8%) and ST7 (11.1%). ST155 and ST1 were respectively detected

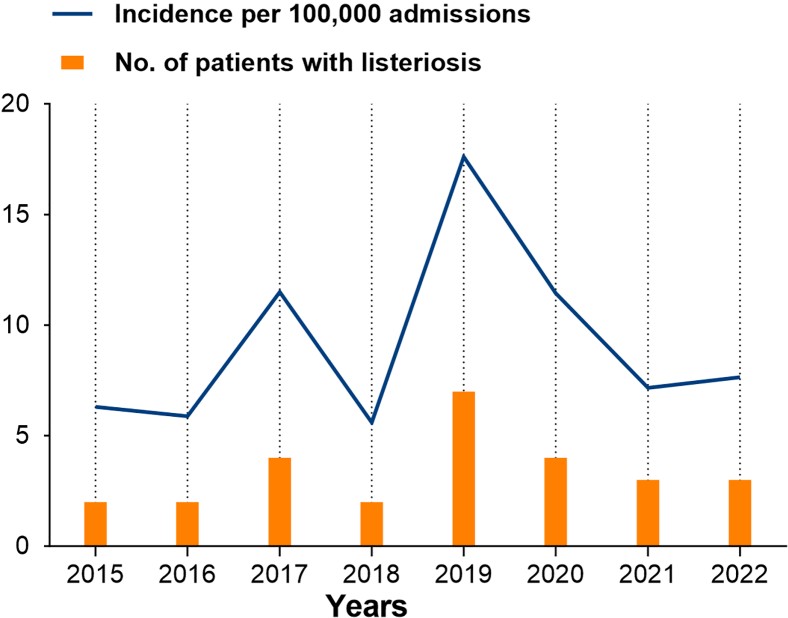

**Figure 2** Annual incidence of listeriosis per 10,000 admissions over the study period in Shandong, China.                                                                 

in 2 (7.4%) strains. Other STs were ST3, ST121, ST5, and ST169 (each $n = 1$). Furthermore, by multiplex PCR-serogroups analysis, the 27 clinical isolates were divided into three serogroups, including 18 (66.7%) serogroup 1/2a, 7 (25.9%) serogroup 1/2b, and 2 (7.4%) serogroup 4b (Table 3). Representative results of multiplex PCR-serogroups analysis are shown in Fig. S1. Among the 11 analyzed virulence genes, ten virulence-associated genes (*hly*, *plcA*, *plcB*, *prfA*, *actA*, *inlA*, *inlB*, *inlC*, *inlJ*, and *iap*) were identified. No strains carried *mpL* gene, whereas *plcA*, *prfA*, *inlA*, *inlJ* genes were detected in all 27 strains. Other commonly detected virulence genes were *InlB* (81.5%), *iap* (63.0%), *hly* (55.6%), *actA* (55.6%) and *InlC* (18.5%).

## DISCUSSION

This study, to the best of our knowledge, is the first to report the incidence of listeriosis in non-perinatal patients from Shandong, China and provide molecular and antimicrobial resistance profiles of *L. monocytogenes* isolates in this population. Our findings highlight the diversity and heterogeneity of this species, with various serogroups and sequence types. Additionally, the reports of treatment and outcomes of these patients and the resistance rate of *L. monocytogenes* isolates provide additional evidence for the management of listeriosis.

In our study, epidemiological data showed an overall incidence of nine cases per 100,000 admissions, which was higher than the historical data reported globally, ranging from 0.3/100,000 to 3/100,000 admissions (*Corral et al., 2024*; *de Noordhout et al., 2014*). This discrepancy may reflect unique factors in China, such as an ageing population and distinct dietary habits. In China, most cases of listeriosis are sporadic, with a similar distribution observed between perinatal and non-perinatal cases (*Fan et al., 2019*; *Feng et al., 2013*; *Li et al., 2018*). Four published reviews have previously reported the number and prevalence

**Table 1 Demographic and clinical characteristics of included patients.**

| Characteristics | Patients ($n$ = 27) |
|---|---|
| Median age (range), years | 56 (8–82) |
| <18 | 5 (18.5) |
| 18–44 | 3 (11.1) |
| 45–59 | 8 (29.6) |
| ≥60 | 11 (40.7) |
| Female sex | 15 (55.6) |
| Clinical symptoms | |
| Fever | 24 (88.9) |
| Abdominal pain | 2 (7.4) |
| Aggravated postoperative pain | 1 (3.7) |
| Comorbidities | 18 (66.7) |
| Hypertension | 9 (33.3) |
| Diabetes mellitus | 6 (22.2) |
| Malignancy | 5 (18.5) |
| Autoimmune disease | 4 (14.8) |
| Hepatitis | 3 (11.1) |
| Coronary heart disease | 2 (7.4) |
| Cerebral infarction | 2 (7.4) |
| Symptomatic epilepsy | 1 (3.7) |
| Department distribution | |
| Critical care medicine | 7 (25.9) |
| Neurology | 6 (22.2) |
| Pediatrics | 5 (18.5) |
| Gastroenterology | 3 (11.1) |
| Oncology | 2 (7.4) |
| Hematology | 2 (7.4) |
| Neurosurgery | 1 (3.7) |
| Osteoarthropathy | 1 (3.7) |
| Laboratory data | |
| Positive blood culture | 13 (48.1) |
| Positive CSF culture | 8 (29.6) |
| Positive blood/CSF culture | 4 (14.8) |
| Positive ascites culture | 1 (3.7) |
| Positive joint fluid culture | 1 (3.7) |
| Treatments | |
| Monotherapy | 10 (37.0) |
| Combination therapy | 16 (59.3) |
| Mortality | 6 (22.2) |

**Note:**
Data are expressed as number (%) or median (range).

**Table 2 Antibiotic resistance profiles of *L. monocytogenes* strains.**

| Antibiotics | Inhibitory zone diameter (mm) | | | Resistant strains (No.) | | | Resistance rate (%) |
|---|---|---|---|---|---|---|---|
| | Sensitive | Intermediate | Resistant | Sensitive | Intermediate | Resistant | |
| Erythromycin | ≥23 | 14–22 | ≤13 | 26 | 0 | 1 | 3.7 |
| Ciprofloxacin | ≥21 | 16–20 | ≤15 | 26 | 0 | 1 | 3.7 |
| Levofloxacin | ≥17 | 14–16 | ≤13 | 27 | 0 | 0 | 0 |
| Gentamicin | ≥15 | 13–14 | ≤12 | 27 | 0 | 0 | 0 |
| Meropenem | ≥16 | 14–15 | ≤13 | 27 | 0 | 0 | 0 |
| Imipenem | ≥16 | 14–15 | ≤13 | 27 | 0 | 0 | 0 |
| Cotrimoxazole | ≥17 | 13–14 | ≤12 | 27 | 0 | 0 | 0 |
| Vancomycin | ≥15 | – | – | 27 | 0 | 0 | 0 |
| Penicillin | ≥20 | – | ≤19 | 27 | 0 | 0 | 0 |
| Ampicillin | ≥20 | – | ≤19 | 27 | 0 | 0 | 0 |
| Amikacin | ≥17 | 15–16 | ≤14 | 27 | 0 | 0 | 0 |

**Table 3 Distribution of lineages, serotypes, and sequence types of *L. monocytogenes* isolates.**

| Lineages | Serogroup | STs | Total isolates |
|---|---|---|---|
| I | 1/2b | ST87 | 4 |
| I | 1/2b | ST3 | 1 |
| I | 1/2b | ST5 | 1 |
| I | 1/2b | ST619 | 1 |
| I | 4b | ST1 | 2 |
| II | 1/2a | ST8 | 12 |
| II | 1/2a | ST7 | 3 |
| II | 1/2a | ST155 | 2 |
| II | 1/2a | ST121 | 1 |
| Total | – | – | 27 |

of listeriosis cases that occurred across different regions in China during various time periods: 147 cases from 1964 to 2010 (*Feng et al., 2013*), 253 cases from 2011 to 2016 (*Li et al., 2018*), 562 cases from 2011 to 2017 (*Fan et al., 2019*), and 759 cases from 2008 to 2017 (*Chen et al., 2020*). Similar to these reports, listeriosis cases were reported every year in our study, with a peak in specific years. According to the historical data before 2017, the prevalence peak of listeriosis occurred around 2014 (*Chen et al., 2020*; *Fan et al., 2019*); our recent data showed a similar tendency, in which the incidence increased between 2015 and 2019, followed by a sharp decline after 2020. Due to regional, period and population-specific differences, sustaining and comprehensive surveillance is urgently needed to understand listeriosis dynamics in China better. Nevertheless, this trend aligns with the pattern reported by *Corral et al. (2024)* during the same period, which may partly be attributed to the heightened public health measures during the COVID-19 pandemic,

reducing the exposure to *L. monocytogenes* or to a shift in surveillance priorities toward monitoring COVID-19. Additionally, the overall incidence was unavailable in these reports due to the lack of comprehensive surveillance; thus, our study reported the data in the recent decade (2015–2022) and added a novel epidemiological perspective by providing annual incidence data. Overall, our findings may help change the current underestimation of this disease in China and highlight the necessity of strengthening epidemiological surveillance.

Because there was no established standard option and duration of therapy for listeriosis, patients herein received empirical therapies based on the clinical condition and clinicians' experience. Consistent with previous practices (*Corral et al., 2024*; *Sutter et al., 2024*), β-lactam antibiotics (*e.g.*, meropenem, ceftriaxone, and ampicillin) are commonly used for our cases. However, we did not observe a clear association of outcomes with treatment regimens or duration, possibly due to the small sample size and heterogeneity of clinical presentations. Despite limited cases, we observed that the majority of patients treated with penicillins were fully cured, which is in line with the international consensus on the priority of penicillin G or ampicillin for the treatment of listeriosis (*Koopmans et al., 2023*). While we used similar regimens, the lack of uniformity in treatment strategies might have contributed to the observed high mortality rate of 22.2%; nevertheless, this rate was comparable to the data in Asia (*Chen et al., 2020*; *Choi et al., 2018*; *Fan et al., 2019*) or Western populations (*Dickstein et al., 2019*; *Jensen et al., 2016*; *Sutter et al., 2024*), underscoring the challenges of treating listeriosis, even with timely and active antibiotic therapy. Our findings highlight the urgent need for further investigation of optimal or alternative therapeutic approaches to improve the clinical outcome and reduce the disease burden, as well as the development of evidence-based treatment guidelines tailored to the Chinese population.

The molecular characteristics observed from our non-perinatal cases are consistent with existing evidence in China (*Cheng et al., 2022*; *Lu et al., 2021*; *Zhang et al., 2019*), with serogroups 1/2a (66.7%) and 1/2b (25.9%) being most prevalent, and ST8 (44.4%) and ST87 (14.8%) as the predominant ST types. Moreover, the distributions of serogroups or MLST types were generally consistent among perinatal and non-perinatal patients, as demonstrated by a surveillance report in China (*Li et al., 2019*), supporting our findings. However, these findings were inconsistent with those in Western countries, where serotype 4b was frequently identified while ST87 was seldom linked to human listeriosis (*Jennison et al., 2017*). It is emphasized that regional surveillance is necessary for this heterogeneous species because specific STs are associated with severe listeriosis cases. Additionally, by reviewing the published literature, there is limited evidence about the virulence characterization of *L. monocytogenes* available, particularly in human listeriosis (*Fan et al., 2019*; *Feng et al., 2013*). In this study, *plcA*, *prfA*, *inlA*, *inlJ* genes were detected in almost all *L. monocytogenes* isolates, and *inlB*, *iap*, *hly*, and *actA* were detected in the majority of isolates. However, the association between virulence genes and pathophysiological mechanisms remain less well characterized and needs an in-depth study. Meanwhile, the favorable antibiotic susceptibility of *L. monocytogenes* isolates is reassuring here, with only two (7.4%) strains resistant to erythromycin or ciprofloxacin. These results are as expected on the basis of prior surveillance (*Yan et al., 2019*; *Zhang et al., 2021*).

Several limitations of our study should be acknowledged. Firstly, selection bias and data integrity due to insufficient records or loss of follow-up are inherent limitations of a retrospective study. Secondly, the small sample size and heterogeneity in included cases limit the generalizability of our findings and preclude a robust analysis of potential correlations between treatments and molecular features or susceptibilities. Thirdly, the data were collected from several regions in China, which cannot fully represent the national epidemiology of listeriosis. In addition, as our data were primarily derived from hospital-based clinical isolates, this may underestimate the prevalence of community-acquired cases and restrict a comprehensive analysis of environmental or foodborne sources. These limitations highlight the need for integrated surveillance systems combining public health, hospital, and food safety to understand listeriosis epidemiology comprehensively and support targeted prevention strategies.

## CONCLUSION

In summary, this study provides valuable insights into the clinical and molecular features of *L. monocytogenes* in non-perinatal patients with listeriosis and adds new information regarding the heterogeneity within this species. Despite low antimicrobial resistance and appropriate treatment, listeriosis remains associated with a significant mortality rate, emphasizing the need for timely diagnosis and effective management strategies, especially in patients with comorbidities. The findings demonstrate that listeriosis remains a significant clinical concern, and molecular surveillance of *L. monocytogenes* strains is critical for understanding the epidemiology of the disease and guiding public health strategies, which may be a way forward in reducing the disease burden. Future studies should focus on larger patient cohorts and explore the potential predisposing factors of this infection.

### Funding

This study was supported by the Jining Key Research and Development Program (Nos. 2022YXNS057, 2023YXNS127 and 2024YXNS028) and Shandong Health Science and Technology Development Program (No. 202211000699). There was no additional external funding received for this study. The funders had no role in study design, data collection and analysis, decision to publish, or preparation of the manuscript.

### Grant Disclosures

The following grant information was disclosed by the authors:
Jining Key Research and Development Program: 2022YXNS057, 2023YXNS127 and 2024YXNS028.
Shandong Health Science and Technology Development Program: 202211000699.

### Competing Interests

The authors declare that they have no competing interests.

## Author Contributions

- Yan Liu conceived and designed the experiments, performed the experiments, analyzed the data, prepared figures and/or tables, authored or reviewed drafts of the article, and approved the final draft.
- Yuan Xie conceived and designed the experiments, performed the experiments, analyzed the data, prepared figures and/or tables, authored or reviewed drafts of the article, and approved the final draft.
- Lei Liu performed the experiments, analyzed the data, prepared figures and/or tables, authored or reviewed drafts of the article, and approved the final draft.
- Jie Wang performed the experiments, prepared figures and/or tables, and approved the final draft.
- Wenjing Li performed the experiments, analyzed the data, prepared figures and/or tables, authored or reviewed drafts of the article, and approved the final draft.
- Chengfan Yang performed the experiments, analyzed the data, authored or reviewed drafts of the article, and approved the final draft.
- Shuhua Lu conceived and designed the experiments, authored or reviewed drafts of the article, and approved the final draft.

## Ethics

The following information was supplied relating to ethical approvals (*i.e.*, approving body and any reference numbers):

This study was approved by the Ethics Committee of the Affiliated Hospital of Jining Medical University (No. 2022-08-C004).

## Data Availability

The raw measurements are available in the Supplemental File.

## Supplemental Information

Supplemental information for this article can be found online at http://dx.doi.org/10.7717/peerj.19126#supplemental-information.

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
