# Peer review of "Clinical, molecular, and resistance features of *Listeria monocytogenes* in non-perinatal patients with listeriosis: 8-year retrospective data from four tertiary hospitals in Shandong, China"

_PeerJ, doi:10.7717/peerj.19126_

## Round 0.1 · original submission · Major Revisions

I am delighted that both reviewers provided a positive response to the manuscript. One of the reviewers highlighted a concern regarding the absence of details about the Institutional Review Board (IRB) approval. I fully agree with the reviewer’s observation about the importance of including IRB details, as it is essential for work of this nature. Below are the points raised by the reviewers that require your attention

Reviewer 1 ·

Basic reporting

The authors use clear and concise language appropriate for the journal. The data tables and figures are clear and easy to understand. There are a few small items that could be changed for added clarity:

Line 59-61: "However, epidemiological surveillance on listeriosis was initiated in a few provinces of China until 2013." I think the connotation based on the context is that surveillance for listeriosis was not widely initiated until 2013. Perhaps rephrasing to "However, epidemiological surveillance of listeriosis was not widely initiated in China until 2013" or something similar would be clearer.

Lines 67-68: "Within this context, urgent efforts are needed to provide more evidence in China." Do the authors mean to say efforts are needed to gather more data? If so, maybe rephrase this sentence to read "urgent efforts are needed to gather more data to better understand the factors affecting treatment and prognosis."

Lines 178-191: Much of this paragraph seems to assert novelty to the author's findings in the context of previously reported literature. I think the discussion would be easier to read if the authors would rephrase it to focus on contextualizing their findings within the spectrum of published data, rather than using snippets of published data to assert the novel aspects of their work. I think rewording this paragraph and comparing/contrasting more clearly with previously published data would assert your novelty without glossing over significant findings as much.

Experimental design

The design is clear to understand and appears to be scientifically rigorous. IRB approval is referenced. I have no specific recommendations for improvement.

Validity of the findings

Line 59: "Over the past decades, the incidence of listeriosis has been increasing worldwide." Please provide a reference to support this assertion. Other studies referenced by the authors do not appear to support a global increase in incidence

Discussion (general): There are some assertions made by the authors that appear contradictory to previously published data that should be expanded upon or at least commented upon more in the discussion.

Lines 184-196: As mentioned above, this portion of the discussion could be rewritten to contextualize the findings within published data better. One specific example: The authors cite Chen et al 2020 as supportive evidence of similar overall mortality rate, but Chen et al did show discrepant incidence over time (peaking in 2014, rather than increasing to 2020 as the author's data shows). I think including discrepant data from these already referenced studies and commenting on possible reasons for these discrepancies would be valuable for the paper.

Lines 208-223: Other studies have performed molecular characterization of listeria isolates in recent years. How does the authors' data compare to the previously reported data and to perinatal patients (i.e. is there evidence that specific serovars preferentially infect certain patient populations)?

The authors mention that no standardized treatment protocol for listeriosis currently exists in China. Does their data allow for any extrapolation to suggest an effective treatment protocol (i.e. were there certain treatment regimens that were associated with lower mortality? Were treatment protocols standardized in other countries used by some of the clinicians?) I think this is important to comment on as the authors do comment on the fact that despite low resistance and antibiotic treatment in most cases, there was still high mortality.

Additional comments

I think this paper adds to a body of knowledge, especially showing region-level differences in incidence for listeriosis. I think the authors could improve this manuscript by correlating their data with the data that is already published on the topic, specifically commenting on how their data differs from others while postulating some reasons as to why these differences exist.

Reviewer 2 ·

Basic reporting

Overall the manuscript is well written for the scope of work intended. Some improvements are required for the manuscript which are:
- Ethics approval was obtained, with the mention of waiver of consent forms. The uploaded file showed the application for waiver of consent form, but not the approval for this waiver. Please clarify if this document showed the approval of ethics committee indicating waiver of the consent form.
- Quality of image for the colonies and b-hemolysis is not clear.
- Label for the colonies A - non-spore forming rod - is not observable on agar plate. Kindly relabel to the appropriate image if needed.
- Figure 1c) I cannot see the synergistic zone and label of which bacteria strains were used.
-Fig 2 - include the province name covered for the epidemiological surveillance in the label
- Since MLST was performed, the analysis and relatedness of the clinical isolates can be presented as lineage to show the prevalence of serious listeriosis strains (an example: Ragon M, Wirth T, Hollandt F, Lavenir R, Lecuit M, et al. (2008) A New Perspective on Listeria monocytogenes Evolution. PLOS Pathogens 4(9): e1000146. https://doi.org/10.1371/journal.ppat.1000146)

- Some analysis on the association of the virulent genes, comorbidities, antibiotic susceptibilities of isolates to the treatment received
- Please revise the citations to be included within the sentences and remove the fullstop before the references in the same sentence

Experimental design

It is a good paper and information especially on surveillance of listeria monocytogenes with some details on microbiological characterization.

Results:

- Please show the Vitek-2/Vitek results
- Does the multiplex PCR results have a positive control Listeria monocytogenes? please state and indicate.

Validity of the findings

Line 194 to 196 - Authors to clarify the: incidence increased between 2015-2019 and decline after 2020 due to reduced exposure or also with the surveillance intended to focus on COVID-19?
Line 199 - Revise: Consistenting to Consistent

- Please state the statistical results

Additional comments

I suggest to include limitation of study:
- to discuss the extent and concern covering this area and potentially tie to which type of surveillance either public health, hospital or food safety

---

## Round 0.2 · accepted · Accept

Dear Dr Lu,

Your manuscript, "Clinical, molecular, and resistance features of Listeria monocytogenes in non-perinatal patients with listeriosis: 8-year retrospective data from four tertiary hospitals in Shandong, China", has now been reviewed and the reviewer comments appended below. Both reviewers are satisfied with your rebuttal and the explanations provided in response to their comments. However, a minor correction has been suggested in line 152 by one of the reviewers. I would appreciate it if you could make the necessary revision before publication.

Congratulations on your publication, and best wishes for your future research.

Thanking you,
Yours sincerely,
Sushanta Deb

Reviewer 1 ·

Basic reporting

The suggested revisions were made and I have no further comments/suggestions to make.

Experimental design

The suggested revisions were made and I have no further comments/suggestions to make.

Validity of the findings

The suggested revisions were made and I have no further comments/suggestions to make.

Additional comments

I commend the authors for their detailed reworking of their discussion to include contextualization of their findings. The final product reads very well and includes the requested revisions.

Reviewer 2 ·

Basic reporting

Improvement to the Literature and structure of manuscript.
In abstract and Introduction.
Listeria monocytogenes causes foodborne illness - uncommonly written as severe infectious disease as the transmission is through food, and not person-to-person.

Experimental design

The translated human ethics study approval included the consent waiver, and the protocols were adhered as per institutional policy and described in the manuscript.

Validity of the findings

Authors have provided rebuttals and clarification on the issues raised.

Figure resolution has been improved for clarity.

Revise: Under Antimicrobial resistance profile and molecular features of L. monocytogenes isolates - Line 152, I suggest to change to "From the antimicrobial susceptibility..."

Additional comments

The discussion has been improved. I understand the limitations of the work and sample size will usually be small due to the incidence of disease, however, in Table 3, perhaps authors can incorporate additional column on clinical outcome, mortality/recovery and co-morbidities it would be a good insight in understanding this disease, even though the sample size is small. This could be beneficial in the future epidemiological studies, from the microbiology and clinical perspective.